# Three-dimensional covariance-map imaging of molecular structure and dynamics on the ultrafast timescale

Jason W. L. Lee[1], Hansjochen Köckert [1], David Heathcote [1], Divya Popat [1], Richard T. Chapman [2], Gabriel Karras[2], Paulina Majchrzak [2], Emma Springate [2] & Claire Vallance [1✉]

Ultrafast laser pump-probe methods allow chemical reactions to be followed in real time, and have provided unprecedented insight into fundamental aspects of chemical reactivity. While evolution of the electronic structure of the system under study is evident from changes in the observed spectral signatures, information on rearrangement of the nuclear framework is generally obtained indirectly. Disentangling contributions to the signal arising from competing photochemical pathways can also be challenging. Here we introduce the new technique of three-dimensional covariance-map Coulomb explosion imaging, which has the potential to provide complete three-dimensional information on molecular structure and dynamics as they evolve in real time during a gas-phase chemical reaction. We present first proof-of-concept data from recent measurements on $CF_3I$. Our approach allows the contributions from competing fragmentation pathways to be isolated and characterised unambiguously, and is a promising route to enabling the recording of 'molecular movies' for a wide variety of gas-phase chemical processes.

---

[1] Department of Chemistry, University of Oxford, Chemistry Research Laboratory, 12 Mansfield Rd, Oxford OX1 3TA, UK. [2] Central Laser Facility, Science and Technology Facilities Council, Rutherford Appleton Laboratory, Harwell Campus, Didcot OX11 0QX, UK. ✉email: claire.vallance@chem.ox.ac.uk

 1

Laser pump-probe experiments have been used for several decades to probe the dynamics of ultra-fast chemical processes, with the achievable time resolution having reached the femtosecond timescale[1] by the mid-1980s. Spectroscopic probing with tens-of-femtosecond time resolution allows the evolving energy level structure of the chemical system under study to be followed in real time as the reaction proceeds, while velocity-map imaging[2,3] (VMI) allows measurement of the evolving reaction product scattering distribution on the same timescale. In a typical VMI measurement, the pump laser initiates the chemical process of interest, the probe laser ionises one or more of the fragments, and the ionised products are then accelerated along a flight tube by an electric field. At the end of the flight tube their velocities are mapped onto a position-sensitive ion detector, and the optical image generated by the ion detector is recorded with a fast camera. The product scattering distribution recorded in this way provides a detailed fingerprint of the forces acting during the reaction, and therefore of the chemical mechanism. Velocity-map imaging studies over the past two decades or so have provided unprecedented insight into the dynamics of a wide variety of unimolecular and bimolecular reactions[4–8], even revealing previously unknown reaction mechanisms[9].

Recently-developed multi-mass velocity-map imaging approaches[10–12] have extended the technique to larger chemical systems by enabling detection of all products on each experimental cycle, an advance over earlier approaches in which only one species could be detected in a given experiment. In addition to substantial gains in data acquisition speed, multi-mass imaging also enables correlations to be uncovered between the scattering distributions of different reaction products, providing even more detailed information on the reaction mechanism. Such correlations are revealed by calculating the statistical covariance[13] between the velocity distributions of reaction product pairs. Covariance analysis[14] has been exploited previously in applications including time-of-flight mass spectrometry[15], photoelectron spectroscopy[16], and gamma ray spectroscopy[17] as a powerful tool for uncovering correlations between the velocities of two or more particles. In the context of velocity-map imaging, the information obtained is similar to that revealed in coincidence imaging measurements[18–22], which have been used to follow reactions in real time[23]. A coincidence experiment records the relative position and arrival times of two or more charged particles arising from the same event by using the arrival of the first (lightest) particle at the detector to trigger recording of signals for the remaining, heavier particles. An event is only stored if all (or at least a predefined minimum number) of the expected particles are detected. Coincidence experiments require a low count rate—typically fewer than one event per experimental cycle—in order to avoid high numbers of false coincidences which would overwhelm the signal from true coincidences. This necessitates a high repetition rate in order to achieve a sufficiently high data acquisition speed to make the experimental measurements feasible. Covariance imaging, in contrast, can be used in much higher count-rate regimes, at either low or high repetition rates, and offers an alternative approach to coincidence imaging, particularly in cases where high repetition rate laser systems are not available. Covariance imaging is likely to prove particularly useful as molecular size increases, since there are no intrinsic limitations on the number of ions that can be detected on each experimental cycle.

In most VMI experiments, the probe laser simply ionises the reaction products, and the VMI measurement provides information on product identities and their energy level structure and scattering distributions. If instead of ionising the products, a probe laser is employed with sufficiently high intensity to initiate Coulomb explosion[13], it becomes possible to obtain structural information directly on the femtosecond timescale, providing an alternative to diffraction methods such as ultrafast X-ray diffraction[24–26] and ultrafast electron-diffraction[27,28]. To initiate a Coulomb explosion, an intense probe laser pulse strips multiple valence electrons from the molecule, eliminating one or more chemical bonds and creating a highly unstable collection of positively charged ions. The nuclei do not have time to move significantly during this process, and are therefore still located at or very close to their original positions within the structure. Coulomb repulsion between the ions rapidly leads to a 'Coulomb explosion', during which the initial positions of the ions are mapped onto their final velocities by the repulsive forces acting between the recoiling ions. This mapping is key to the structural measurement. By measuring the ion velocities in the velocity-map imaging step, one can in effect 'work backwards' to the original molecular structure. Under ideal conditions the probe laser would strip a sufficient number of electrons from the sample molecule to cause it to explode into atomic ions, the so-called 'pure Coulomb explosion' regime. We have demonstrated this approach in our previous work on substituted methanes, bromines, and biphenyl molecules[13,29–32]. These earlier experiments employed traditional 'crushed' velocity-map imaging, and we were therefore limited to recording two-dimensional projections of the velocity distributions. Using two-body and three-body covariance approaches, these were analysed to obtain two-dimensional projections of the molecular structures, to distinguish between structural isomers, including enantiomers, and to follow structural change on the femtosecond timescale.

In this work, we extend the approach to three-dimensions, demonstrating 3D-sliced[33,34] Coulomb-explosion covariance-map imaging for the first time. For this first demonstration we focus on the one-laser photoionization and subsequent fragmentation and Coulomb explosion of $CF_3I$. As well as demonstrating the potential for structural studies, we also show that covariance-map imaging offers a powerful tool for disentangling contributions to the overall signal from competing reaction pathways. In particular, it shows definitively which reaction products are (and are not) formed together during a chemical event, since covariances will only be seen between these products. Such tools will become increasingly important as the field of reaction dynamics moves away from studies of small molecules and towards investigations into larger, more complex chemical systems.

## Results

**Identification of reaction products**. As explained in the Introduction and Methods, following irradiation of gas-phase $CF_3I$ with a high intensity, ultra-short 800 nm laser pulse, the imaging detection system records the position and arrival time of each ionised fragment at the position-sensitive detector. Integrating over the spatial coordinates yields the time-of-flight spectrum, while plotting the spatial coordinates over the range of arrival times corresponding to a chosen reaction product reveals the scattering distribution for that species. The time-of-flight mass spectrum for the charged fragments formed following irradiation of neutral $CF_3I$ by a 535 μJ, ~40 fs pulse from the 800 nm Artemis laser is shown in Fig. 1. We note that the detection sensitivity of the ion detector was attenuated at the arrival times of very intense peaks corresponding to $m/z = 196$ (parent ion), 98 (doubly-charged parent ion), and 18 ($H_2O$) in order to prevent damage to the ion detector. Time-gating of the $H_2O$ background signal in this way unfortunately reduced our ability to detect the nearby $F^+$ ion, with $m/z = 19$. Signal from the atomic ions $C^+$, $I^+$, $F^{2+}$, $C^{2+}$, $I^{2+}$ and $I^{3+}$ is clearly visible in the mass spectrum, as is significant signal from molecular fragment ions, including $CF_3^+$, $CF_2^+$, $CF_3^{2+}$, $CF^+$ and $CF_3^{3+}$, indicating that sufficient ionisation was achieved

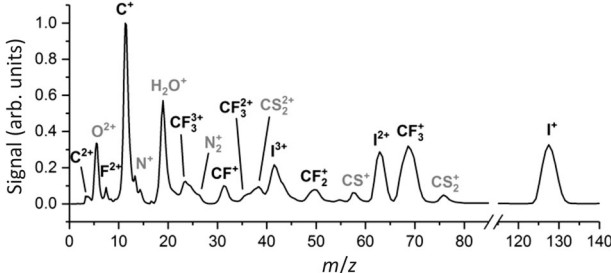

**Fig. 1 Reaction product identification.** Charged fragments are identified from their arrival times within the time-of-flight spectrum recorded following laser-induced Coulomb explosion of $CF_3I$. Peaks of interest are labelled in black, with background peaks labelled in grey.

to initiate Coulomb explosion and generate a variety of atomic and molecular fragment ions. Note that there is some signal ($O^{2+}$, $N_2^+$, $H_2O^+$) from residual air in the chamber, and some ($CS_2^{2+}$, $CS^+$, $CS_2^+$) from a previous sample that could not be completely eliminated from the system within the available experimental time. However, these peaks are well separated in time from the $CF_3I$ peaks of interest, and with the exception of $H_2O^+$ discussed above, do not interfere with the signal.

**Observed fragmentation pathways from covariance in two-dimensions.** Fragmentation of doubly-charged $CF_3I^{2+}$ ions following double photoionisation of neutral $CF_3I$ has been studied previously[35–37]. Four main product channels have been identified, yielding the products $CF_3^+ + I^+$, $CF_2^+ + I^+ + F$, $CF^+ + I^+ + 2F$ and $CF_2I^{2+} + F$, respectively. The time-of-flight mass spectrum shown in Fig. 1 reveals the presence of most of the ionic products from these processes, but does not in itself allow us to determine whether they are formed via the processes above, or via fragmentation of more highly charged parent ions. The presence and subsequent fragmentation of parent ions with more than two charges is clear from the presence of multiply charged fragment ions (e.g., $C^{2+}$, $F^{2+}$, $CF_3^{2+}$, $CF_3^{3+}$, $I^{2+}$, $I^{3+}$) in the mass spectrum. Fragmentation of highly charged $CF_3I$ ions has been studied previously by Douglas[38]. Our intention in this work is not to carry out an exhaustive dynamical study of the photo-fragmentation dynamics of multiply charged $CF_3I$, but to use $CF_3I$ as a test system to demonstrate the way in which covariance-map imaging can be used to disentangle complex signals arising from multiple competing dissociating pathways. We will do this with the aid of a few examples.

We begin by considering the dominant and lowest energy Coulomb explosion processes observed under our experimental conditions (and previously by others[35–38]), which involve cleavage of the C–I bond following formation of a multiply charged parent ion, i.e.,

$$CF_3I^{n+} \rightarrow CF_3^{m+} + I^{(n-m)+} \tag{1}$$

In terms of a strategy for identifying the various possible product pairs from the velocity-map images, we note that each charge-state pair experiences a characteristic Coulomb repulsion between the recoiling fragments, yielding fragment velocity distributions that are specific to each channel. For example, the measured velocity distribution for $CF_3^+$ ions contains contributions from channels involving I, $I^+$ and $I^{2+}$ partner fragments, which manifest in the $CF_3^+$ velocity distribution as a series of concentric spheres. These rings are seen clearly in the full 3D-sliced velocity-distribution for $CF_3^+$ shown on the left-hand side of Fig. 2, and also in the central slice through the distribution shown on the top right-hand side of the same figure.

Disentangling all of the competing fragmentation pathways that contribute to a particular ion signal in a velocity-map imaging experiment has in the past been a complex process. However, covariance analysis of the recorded data offers a vast simplification. Since covariances are only observed between ions that are formed in the same event, transforming the measured velocity-map images into covariance images for specified ion pairs allows the contribution to the overall ion signals from each pathway to be identified unambiguously. Covariance-map images are shown for the channels $CF_3^+ + I^+$ and $CF_3^+ + I^{2+}$ on the right hand side of Fig. 2, reduced to 2D projections for visualisation purposes. The central slices of the full 3D scattering distributions for the $CF_3^+$, $I^+$ and $I^{2+}$ fragments are shown in the top panel, with the covariance images between the various ion pairs plotted below. In each covariance image, one of the two products is chosen as the reference ion, travelling in the reference direction towards the right of the image, indicated by an arrow. The intensity within the image shows the velocity distribution of the partner fragment relative to the reference ion. In each case, a clear covariance signal is seen, revealing—as expected for a pairwise dissociation—that the two fragments recoil in opposite directions with well-defined velocities. Some residual background signal is also present in other regions of the covariance images. This arises from imperfect subtraction of 'false covariances' during the conversion from velocity-map images to covariance-map images (see Methods section), and reduces in intensity as the number of experimental cycles included in the data set is increased. This background signal can safely be ignored for the purposes of the present discussion.

As noted above, the final velocities of the fragments are determined primarily by the total charge on the dissociating parent ion, which determines the strength of the Coulomb repulsion between the $CF_3^+$ ion and its $I^+$ or $I^{2+}$ partner ion. The repulsion is higher in the case of more highly charged ions, yielding higher velocities for the $CF_3^+ + I^{2+}$ channel than for the $CF_3^+ + I^+$ channel. This is seen clearly in the covariance-map images, with the $CF_3^+$ velocity peaking further from the centre of the image when partnered with $I^{2+}$ than when partnered with $I^+$. The total kinetic energies of the $CF_3^+$ and $I^+/I^{2+}$ ions formed in the two channels are found to be 5.6 eV and 10.55 eV. Based on the 2.144 Å equilibrium bond length of the C–I bond in $CF_3I$, these energies are significantly lower than would be expected if Coulomb repulsion were the only force acting on the departing ions. In common with the findings of other authors[39–42], we note that all relevant chemical forces must be included in order to construct an accurate model of the potential energy surface over which the dissociation proceeds.

As a second example, Fig. 3 shows the central slice of the scattering distribution recorded for $CF_2^+$, together with the covariance-map image for $CF_2^+$ formed together with $I^+$. While there are various possible routes to forming the $CF_2^+ + I^+$ ion pair from multiply-charged $CF_3I$ ions, we believe that the form of the covariance image implies that they arise from dissociation of $CF_3I^{2+}$ to form $CF_2^+ + I^+ + F$. Eland et al. showed[35] that this is the next lowest energy fragmentation pathway for $CF_3I^{2+}$ after the $CF_3^+ + I^+$ pathway considered above. The ion pair is formed in a two-step mechanism: the $CF_3I^{2+}$ parent ion first loses a neutral F atom, leaving behind $CF_2I^{2+}$, which subsequently decays into $CF_2^+ + I^+$. The detailed form of the covariance image can be rationalised by considering the forces acting in each of the two steps. The image is very similar to those observed in Fig. 2, which resulted from axial recoil of $CF_3^+$ from $I^+$ or $I^{2+}$. However, the covariance is not as sharp. Loss of neutral F from $CF_3I^{2+}$ results in a small momentum 'kick' to the resulting $CF_2I^{2+}$ intermediate. Though the forces involved in this step are much

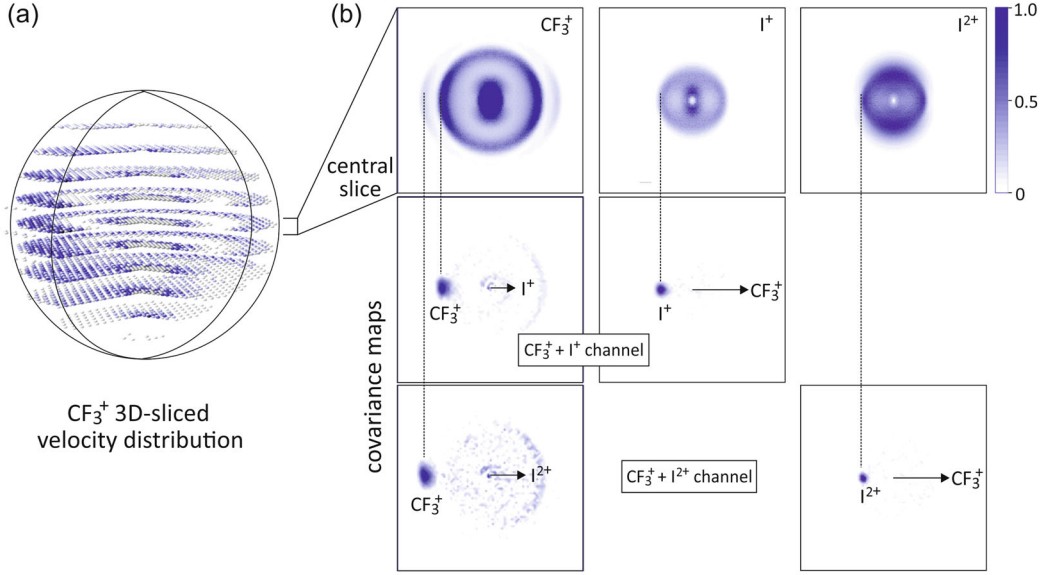

**Fig. 2 3D and 2D velocity-map and covariance-map images. a** 3D-sliced velocity distribution recorded for $CF_3^+$ fragments; **b** central slice of the 3D scattering distributions for the $CF_3^+$, $I^+$ and $I^{2+}$ products. Below these are shown the covariance-map images for $CF_3^+$ formed with an $I^+$ partner ion and with an $I^{2+}$ partner ion. The vertical dotted lines show which features of the scattering distributions correspond to the pathways revealed in each of the covariance signals. Note that the signals in the centres of the distributions correspond to ions formed with a neutral partner, which do not give rise to a covariance signal since the neutral partners cannot be detected in our experiment. All images are plotted on a normalised intensity scale.

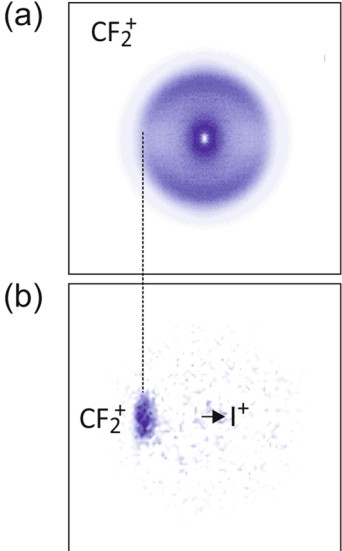

**Fig. 3 2D velocity-map and covariance map images for the $CF_2^+ + I^+ + F$ channel. a** central slice of the 3D scattering distribution for the $CF_2^+$ product of $CF_3I^{2+}$ dissociation; **b** covariance-map image of $CF_2^+$ formed with an $I^+$ partner ion. Both images are plotted on a normalised intensity scale.

smaller than the Coulomb repulsion acting in the second step, they are sufficient to cause a small blurring in the covariance between the $CF_2^+$ and $I^+$ fragments.

**Covariance in three dimensions**. We now move on to considering the data set in three dimensions in order to demonstrate 3D-sliced covariance-map imaging. As an example, we will focus our attention on the $CF_3I^{2+} \rightarrow CF_3^+ + I^+$ fragmentation channel. To visualise the covariances in three dimensions, the covariance analysis is carried out for each pair of slices within the 3D-sliced data sets for $CF_3^+$ and $I^+$. A subset of

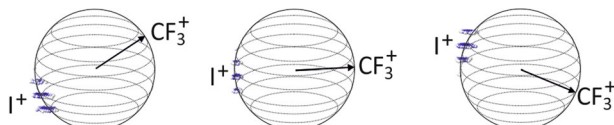

**Fig. 4 3D covariance-map images for the $CF_3^+ + I^+$ fragmentation channel.** The direction of travel of the $CF_3^+$ reference ion is shown by the black arrow; the velocity distribution of the $I^+$ ion relative to this reference direction is shown by the intensity distribution within each slice of the 3D covariance-map.

the results of this analysis are shown in Fig. 4. In each of the three panels, $CF_3^+$ ions within a single slice are used to define the reference direction, and covariances with all slices of the $I^+$ data set are plotted. As was the case in two-dimensions, the $I^+$ ions are seen to recoil in a direction opposite to that of the $CF_3^+$ reference ion. Though the recoil velocity is relatively well defined, as is the angular distribution within the plane of any single slice, the distribution is considerably broader than might be expected in the direction defined by the time-of-flight axis. Assuming axial recoil of the two fragments, and performing a Gaussian fit to the angular distribution within the covariance map images, we estimate the full-width-half-maximum angular resolution to be approximately 6 degrees in the azimuthal angle (within the slicing plane) and 25 degrees in the polar angle (measured from the time-of-flight axis). We attribute the observed blurring of the velocity distribution along the time-of-flight axis to a combination of factors, including the laser pulse length, a small amount of space-charge repulsion within the expanding ion cloud arising from ionisation of the He molecular beam carrier gas, the decay lifetime of the phosphor screen that forms part of the ion detection system, and small imperfections in the velocity-mapping field. It will be possible to improve on all of these in future experiments. The 3D covariance analysis can be repeated for all pairs of fragments in order to unpick the fragmentation dynamics associated with each fragmentation channel in three dimensions.

## Discussion

We have introduced the technique of 3D-sliced covariance-map imaging as a new probe of molecular structure and dynamics, and have demonstrated its use for the first time in a set of measurements to investigate the laser-induced Coulomb explosion dynamics of $CF_3I$. We have shown that the approach is able to identify individual contributions to a signal comprising contributions from many different competing fragmentation pathways, providing a powerful tool for disentangling complex dynamics within large molecular systems. While this study illustrates this in the context of dissociative photoionization, the method is equally applicable to studying any photoinitiated process when an appropriate pump-probe laser scheme is employed to initiate reaction and ionise the products to be detected.

Of equal if not greater interest is the potential to use covariance-map imaging in combination with molecular Coulomb explosions as a probe of molecular structure in the gas phase. We have shown previously[13,24,30–32] that significant structural information can be obtained when employing this approach in conventional two-dimensional 'crushed' velocity-map imaging approaches, and in this work we extend the experimental capabilities to three dimensions by incorporating slice imaging into the velocity-map imaging detection scheme. This enables us to visualise covariances in three dimensions for the first time. We note that structural information is obtained on the femtosecond timescale, with no need for preparing spatially aligned or oriented reactants, and no need for image inversion algorithms.

In this work, the laser pulse energy available at the Artemis ultrafast laser facility was in the range yielding both atomic and molecular fragment ions. With higher laser pulse energies, such that Coulomb explosion into atomic ions is achieved, the approach is able to provide direct information on three-dimensional molecular structures via the mapping from atomic positions to fragment ion velocities that occurs during the Coulomb explosion. Achieving this requires that the mapping is well understood. Previously, a variety of wavepacket and trajectory-based approaches have been applied to the problem[24,39–42], in many cases with a high degree of success. Within our own group, we are currently developing on-the-fly/Born-Oppenheimer molecular dynamics trajectory-based methods which take into account all of the chemical forces acting on the separating fragments during a Coulomb explosion. The eventual aim is to implement a 'structure refinement' algorithm to obtain molecular structures from covariance-map images given an 'initial guess' structure, in an analogous way to structure refinement of crystalline solids from X-ray diffraction data.

3D-sliced covariance-map imaging is currently at a relatively early stage of implementation, but with further development will enable recording of three-dimensional 'molecular movies' in real time during a reactive event.

## Methods

**Coulomb-explosion velocity-map imaging measurements**. The experiments were performed at the Artemis ultra-fast laser facility, part of the Central Laser Facility in Harwell, Oxfordshire[43]. The Artemis laser is a customised and modified RedDragon 1 kHz Ti:Sapphire laser from KMLabs, operating at 800 nm.

The velocity-map imaging experiment is contained within a high-vacuum system with a base pressure of around $10^{-7}$ mbar. Neutral $CF_3I$ was prepared in a highly diluted seeded molecular beam with He as the buffer gas at a stagnation pressure of 1.2 bar, via supersonic expansion through a Parker Hannifin Series 9 General Valve operating at a repetition rate of 20 Hz. After passing through a skimmer into the velocity-map imaging lens, the molecular beam was intersected by a 535 μJ, ~40 fs pulse from the Artemis laser, initiating multiple ionisation and Coulomb explosion. In addition to employing a very dilute gas mixture, the molecular beam intensity within the vacuum chamber was carefully controlled in order to minimise space-charge repulsion: a molecular beam intensity that is too high results in considerable distortions to the velocity-map images, with a highly detrimental effect on velocity resolution.

The velocities of the ions generated during the interaction between the laser and molecular beam are measured using 3D-sliced velocity-map imaging. The electric field maintained within the velocity-map imaging lens is used to extract the ions along a time-of-flight tube towards a 75 mm diameter position-sensitive detector (Photonis), separating the ions by arrival time according to their mass-to-charge ratio. The velocity-map imaging lens was based on the five-lens-element 'DC sliced imaging' design of Townsend et al.[34], and the electric field within the lens is tuned to retain the three-dimensional structure of the scattering (velocity) distribution for each of the ions following the procedure described by the same authors. Typical potentials applied to the five lens elements were 4000 V, 3560 V, 2400 V, 1600 V and 0 V. The final lens element is always grounded to maintain field-free conditions within the flight tube. The position along the axis at which the laser beam crosses the molecular beam within the VMI lens is not critical; different positions simply result in slightly different optimum slicing potentials.

The detector is comprised of a pair of microchannel plates (MCPs) coupled to a fast phosphor screen. Each ion striking the front face of the detector generates a small spot of light on the phosphor, allowing the scattering distribution of each ion to be visualised as the ions strike the detector. The potentials applied to the MCPs were time-gated to reduce the signal intensities from highly abundant ions ($He^+$, $H_2O^+$ and $N_2^+$) in order to avoid damage to the detector. Under typical operating conditions, the front MCP was grounded, and the back MCP was switched between 500 V ("off") and 1850 V ("on"). The phosphor was held at a potential of 4500 V. A PImMS multimass imaging camera[11,12] positioned behind the phosphor screen records the arrival time and position of each ion with a time resolution of 12.5 ns. The resulting data set comprises images of the three-dimensional product velocity distribution for each mass-to-charge ratio.

At low laser powers, no Coulomb explosion was observed. As the laser power was increased up to the optimum value of 535 μJ, the formation of higher charge states of the parent ion could be inferred from the appearance of increasingly high charge states of atomic iodine in the time-of-flight spectrum, together with the appearance of additional high-velocity rings in the $CF_3^+$ images. When optimising the experimental conditions, the delay between the laser pulse and camera trigger can be tuned to ensure that the centre slice of the velocity distribution is captured by maximising the radius of the observed image for a fragment of interest. This is useful when comparing the results of 3D slice imaging with conventional 2D slice imaging experiments.

**Data processing and analysis**. Conversion from pixel number to velocity was achieved using a calibration based on measurement of Coulomb-explosion images for the well-characterised Coulomb explosion products of $N_2$ in various charge states[44], and confirmed using SIMION 8.0 ion trajectory simulations[45]. The raw images were Abel inverted to determine the velocity distribution of each ion species, which is related to the initial position of the ion within the probed molecular structure via the forces experienced during the Coulomb explosion.

Correlated velocity-distributions between pairs of ions were obtained through a covariance analysis of the data. This process has been described in detail previously in the context of conventional 'crushed' velocity-map imaging, in which the recorded images represent two-dimensional projections of the full three-dimensional velocity distribution[13,24,30–32]. The covariance between two parameters A and B is defined as the average of the product of their deviations from their respective mean values $\langle A \rangle$ and $\langle B \rangle$.

$$\mathrm{cov}(A, B) = \langle (A - \langle A \rangle)(B - \langle B \rangle) \rangle = \langle AB \rangle - \langle A \rangle \langle B \rangle \qquad (2)$$

When applied to the variation in signal within each pixel of the velocity-map images for two different ions, one ionic species is designated as the 'reference' ion, and a second species is designated as the 'signal' ion. The covariance is calculated between each pixel in the reference image and each pixel in the signal image, with the averaging carried out over the number of experimental cycles in the acquisition. At the end of this process, each pixel in the reference image has an associated covariance map which shows the covariance between this pixel and all of the pixels in the signal image. To generate the overall covariance image between the two ions, each individual covariance map is rotated such that the reference pixels lie along a common vector, and the maps for the individual reference pixels are summed.

In this work, the 2D covariance maps were generated from the central slices of the full three-dimensional data set for the reference and signal ions, while the 3D covariance maps were generated by calculating the covariances between all individual pairs of slices of the measured scattering distributions for the two ions.

## Data availability
All relevant data are available from the authors on request.

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

## Acknowledgements
We would like to thank Phil Rice, Alistair Cox and Dave Rose, technical staff at the Artemis facility, the Electronic and Mechanical workshop staff at the University of Oxford's Department of Chemistry, and all members of the PImMS (Pixel Imaging Mass Spectrometry) collaboration (https://pimms.chem.ox.ac.uk). We are very grateful for access to the Artemis facility from STFC and for financial support from EPSRC Programme Grant EP/L005913/1.

## Author contributions
C.V. devised the experiments. C.V., J.W.L.L., H.K., D.H., D.P., R.C., E.S., G.K. and P.M. performed the experiments. J.W.L.L. and C.V. analysed the data and prepared the paper.

## Competing interests
The authors declare no competing interests.
