## [Peer Review File · Communications Chemistry]

Reviewers' comments:

Reviewer #1 (Remarks to the Author):

The manuscript written by Lee et al. presents a new paradigm of ion imaging: 3D-covariance Coulomb explosion imaging as a new tool for instantaneous molecular structures. All the ions ejected upon femtosecond laser pulse irradiation were imaged with a position- and time-sensitive camera, and 3D covariance analyses was done for some group of ionic species. Molecular structural information just before the Coulomb explosion can be reconstructed from the observed ion covariance patterns. The authors carried out proof-of-principle experiments on the Coulomb explosion process of CF₃I molecules, and succeeded in obtaining well-resolved two-body three-dimensional covariance map.

The present experimental achievement opened a new way to track molecular movies of various species. Although there are several overstatements and sentences to be corrected, I think this article is publishable in Comm. Chem. after the following comments are seriously considered.

1. I believe that they substantially overstate the influence of 3D-covariance imaging. I agree that Coulomb explosion imaging (CEI) is a complementary way to diffraction-based method. However, for CEI, 3D-coincidence measurements have been a powerful and universal approach. The authors did not mention the previous reports of coincidence imaging although some of the molecular movie measurements and/or real-time chemical reaction tracking were already reported. (e.g. Hishikawa et al., Physical Review Letters 99: 258302, 2007). Moreover, In page 1, 3rd paragraph, they claimed 3D-sliced imaging is a novel method for structural studies without mentioning anything about Coulomb explosion. To determine the molecular structure, all the atomic components should be imaged as they mentioned later. I strongly recommend the authors to respect Coulomb explosion itself and 3D coincidence imaging.
2. Related to the background of gas-phase imaging, some statements are misleading. For example, ref.2 and 3 are referred as VMI studies, however, ref 2 is for ion imaging.
3. In page 4 the last paragraph, I could not follow their explanation on Fig.3 well. Probably "I⁺" and "CF₃⁺" were swapped in some points, making the manuscript complicated.
4. In Fig.2 bottom, some "residual signal" is clearly seen. They claimed this can be ignored. I agree their claim, but I think its better to add the plausible origin of the background to indicate potential applicability of the present method. I also think it would be more helpful to the reader if the partner I⁺ and/or I₂⁺ images are shown.
5. Is it possible to define the angular (steradian) or velocity (XY, XZ, XYZ) resolution? I think the present method has higher positional resolution but lower temporal resolution relative to the delay-line-based method. In the present manuscript, only time-resolution was given. Because they present the method as a new tool for dynamical study, I think it is better to mention the resolution.

Reviewer #2 (Remarks to the Author):

This paper takes the study of CF₃I ion dissociation as an example to demonstrate the technical advantages of the new method of combining ultrafast laser-pumped probe method and three-dimensional covariance Coulombic explosion imaging, which has the potential to provide molecular structure and dissociation dynamics information as well as the competitive cleavage pathways. Briefly, the authors applied 3D slice imaging to the central slice with CF₃⁺ to record the velocity distributions of the fragment in three dimensions, and then performed covariance analysis of the data to uncover correlations between the velocities of two or more fragments.

The point of this paper is clear, but the results and discussion sections are too simple to understand what the data are all about. The previous studies, such as reference 20 and 21, have

shown that the ultrafast laser pump-probe method coupled with covariance 2-Dimensional imaging of Coulomb explosion can indeed provide structure and dynamics information on molecular photodissociation. This study has improved the above method from 2-Dimensional velocity-map imaging to 3-dimensional slice imaging, based on the "DC sliced imaging" lens design. However, the dissociation dynamics of CF₃I²⁺ has not been clearly explained with the new method. The paper needs to add more details about the study of the dissociation of CF₃I molecules or ions, such as:

- (1) How the image of CF₃⁺ and I⁺ change with the delay time between the pump and the probe laser;
- (2) How the laser energy and laser path affect slice imaging and How to get the center slice in the Coulomb explosion imaging?
- (3) What structure information can the image provide?
- (4) How to separate each channel on the total ion signal? What kinds of dissociation dynamics are involved?

Although some of the work mentioned in the paper is still in progress: "Work is currently underway to develop trajectory simulation software that will provide the link between the measured images and the molecular structure, with the eventual aim of implementing a 'structure refinement' algorithm to obtain structures from covariance-map images in an analogous way to structure refinement of solids from X-ray diffraction data.", I suggest considering publication after adding more technique details and data results and discussions to clarify the related dissociation process. The journal name should be added in reference 21.

Reviewer #3 (Remarks to the Author):

This paper proposes what could be a powerful method for molecular structure determination, and I believe that this idea should definitely be published and would have considerable impact. However, in its present form the proof is too limited, the data contain no quantitative information and it is impossible to assess the attainable precision and the range of applicability of the technique. More data, quantitative results and further discussion would be required. These are my main comments:

1. The authors have concentrated on the straightforward two-body breakup CF₃ – I and present nothing on the less obvious routes producing for instance F₂⁺. For a new technique, it would be essential to prove its validity beyond what is attainable through more conventional methods.
2. Even for the processes the authors have concentrated on (CF₃⁺ + I⁺ and CF₃⁺ + I₂⁺), the data shown are not quantitative, it is not possible to even estimate the degree of blurring of the velocity distributions and thence the attainable precision. All that is graphically shown is momentum conservation. This should be resolved.
3. In the (extremely short!) Discussion section, the authors say "With higher laser pulse energies, such that Coulomb explosion into atomic ions is achieved, the approach is able to provide direct information on three-dimensional molecular structures via the mapping from atomic positions to fragment ion velocities that occurs during the Coulomb explosion.". This is a very interesting proposition but it remains to be demonstrated. One would first have to demonstrate that one can realistically break a polyatomic system into atomic ions and indicate the regime where this would happen (intense XFEL radiation?). References/previous work on this should be added. Also, the authors should give some indications on how they envisage the application of the covariance analysis in this multi-body situation (and how they would deal with the problem mentioned in comment 4 below).
4. On page 4 the authors say "These energies match the Coulomb potential energy of the relevant charges initially separated by a distance around 20% greater than the 2.144 Å C-I bond distance in neutral CF₃I, implying that the charge on the CF₃⁺ fragment is somewhat delocalised over the whole ion rather than being localised on the carbon atom.". In my view this is a superficial

hypothesis for a rather complex problem. There is ample previous literature on this general finding that observed kinetic energies resulting from Coulomb explosion processes are lower than those expected from Coulomb repulsion at equilibrium distances. There has been considerable effort to understand this, through different approaches (charge-resonance-enhanced ionization, ionization ignition, deviations from coulomb potentials,...). The authors should revise these, cite the relevant papers and discuss this effect properly. I include some of the relevant papers on this matter: T. Zuo et al., Phys. Rev. A, 52, R2511 (1995); C. Rose-Petruck, et al., Phys. Rev. A: At.Mol. Opt. Phys., 55, 118 (1997); H. Liu et al., J. Chem. Phys. 126, 044316 (2007); M.E. Corrales, J. Phys. Chem. A 116, 2669 (2012). Also, the authors should indicate a route to extract structural information given this discrepancy with Coulomb energies expected for point charges at equilibrium distances. More generally, the Discussion section should be lengthened according to the ideas of comments 3 and 4 and an outlook for the applicability of the technique to larger systems should be provided.

5. I also believe that the paper should give proper credit to previous work on this matter. The 30-year old work by Vager et al., Science 244, 426 (1989) is a good example, but there are many other important ones (work by Stapelfeldt, Corkum, etc.)

In addition to these general comments, I also have some comments on the presentation of results and method:

- Figure 1 in page 3. The mass spectrum is extremely congested with spurious signals (CS₂, H₂O, N₂). The authors should either provide an explanation for this / explain why it has not been possible to clear the spectrum out, or redo the experiment in cleaner conditions.
- It is necessary to provide further information on the pulsed valve. The conditions of the molecular beam expansion and its repetition rate are not clear.
- The paper would notably improve with more information regarding ion extraction conditions. The reader should have the relevant information for the slicing procedure, that is, the voltages applied and the definition of "slice", that is, how the data are time-binned. The authors say "The electric field is tuned to retain the three-dimensional structure of the scattering (velocity) distribution for each of the ions", but they do not explain how this "tuning" is performed.
- There is also some lack of detail on gated voltages applied. In the description of the mass spectra the authors seem to indicate that the detector gain was turned down at the arrival times of the most intense peaks. The authors should indicate what degree of attenuation was employed, and with what time resolution. Also, I would expect a discontinuity in the MS shown in Figure 1 if the gain for H₂O is attenuated. Why is that not visible?
- The authors should at least briefly describe the covariance analysis method as they apply it to their situation, even if they provide the relevant previous papers.

Response to referees

We would like to thank the referees for their careful reading of the manuscript and for their thoughtful suggestions for improvements. We have now implemented all of these suggestions, as detailed below. We have made very substantial changes to the manuscript, to the point where we highlighting every change within the manuscript file itself would not be particularly helpful. We therefore hope that the following description of our changes will be sufficient for the editors to use as the basis for a decision on publication. The numbering in our response below follows the numbering in the referees' reports.

Reviewer #1

1. *"I believe that they substantially overstate the influence of 3D-covariance imaging. I agree that Coulomb explosion imaging (CEI) is a complementary way to diffraction-based method. However, for CEI, 3D-coincidence measurements have been a powerful and universal approach. The authors did not mention the previous reports of coincidence imaging although some of the molecular movie measurements and/or real-time chemical reaction tracking were already reported. (e.g. Hishikawa et al., Physical Review Letters 99: 258302, 2007). Moreover, In page 1, 3rd paragraph, they claimed 3D-sliced imaging is a novel method for structural studies without mentioning anything about Coulomb explosion. To determine the molecular structure, all the atomic components should be imaged as they mentioned later. I strongly recommend the authors to respect Coulomb explosion itself and 3D coincidence imaging."*

We have added a discussion of coincidence imaging to the Introduction, explaining the experimental regimes under which coincidence and covariance approaches can be used, and including appropriate citations. We have also clarified the reference to obtaining structural information from Coulomb explosions and reordered some of the description within the introduction in response to this comment.

2. *"Related to the background of gas-phase imaging, some statements are misleading. For example, ref.2 and 3 are referred as VMI studies, however, ref 2 is for ion imaging."*

We do not agree with this comment about the use of inappropriate citations for the velocity-map imaging (VMI) technique. The citations we included are the two standard references for the VMI technique, and we and others cite these in all of our work employing the technique. Ion imaging is the technique on which VMI is based; VMI simply offers improved resolution as a result of a small change in the design of the ion lens used to effect the mapping from the ion formation region onto the position sensitive detector. We did not intend these references to be representative of the vast number of VMI studies that have been performed to date, just to acknowledge the inventors of the method appropriately.

3. *"In page 4 the last paragraph, I could not follow their explanation on Fig.3 well. Probably "I+" and "CF3+" were swapped in some points, making the manuscript complicated."*

The referee is correct that there was an error in our labelling of the covariance images. For clarity, we have changed Figure 2 so that it now includes all covariances between the CF_3^+ , I^+ , and I^{2+} Coulomb explosion products, and have checked (multiple times!) that these are all correctly and consistently described in the text and Figure caption. Note that this also addresses a request made in comment number 4 of the referee's report.

4. *"In Fig.2 bottom, some "residual signal" is clearly seen. They claimed this can be ignored. I agree their claim, but I think its better to add the plausible origin of the background to indicate potential applicability of the present method. I also think it would be more helpful to the reader if the partner I+ and/or I2+ images are shown."*

We have provided further description and discussion of the background signal observed in the covariance images. This arises from false covariances within the data set, and is well understood.

5. *"Is it possible to define the angular (steradian) or velocity (XY,XZ, XYZ) resolution? I think the present method has higher positional resolution but lower temporal resolution relative to the delay-line-based method. In the present manuscript, only time-resolution was given. Because they present the method as a new tool for dynamical study, I think it is better to mention the resolution."*

We have now included a discussion and quantification of the angular resolution of the 3D covariance mapping technique, and ways in which this will be improved in future experiments.

Reviewer #2

"The paper needs to add more details about the study of the dissociation of CF₃I molecules or ions."

We have provided additional details of the dissociation dynamics of multiply charged CF₃I ions, in the context of previous work reported in the literature. In response to the referee's specific numbered comments.

1. *"How the image of CF₃⁺ and I⁺ change with the delay time between the pump and the probe laser"*

The referee appears to have misunderstood the nature of our experiment. This was a single-laser experiment in which multiply-charged CF₃I ions were generated by the interaction between the laser and a molecular beam, and the ions subsequently dissociated spontaneously to yield the various fragment ions detected in our measurements. There was no separate probe laser, and therefore no pump-probe delay that could be adjusted. We have tried to clarify our description of the experiment in the hope of precluding such misunderstandings for other readers.

2. *"How the laser energy and laser path affect slice imaging and How to get the center slice in the Coulomb explosion imaging?"*

In the Methods section, we have now explained how the laser energy effects the observed signal. We have also explained that the laser path has little effect so long as the laser crosses the molecular beam, but that translating the laser beam along the time-of-flight axis direction does affect the velocity-mapping potentials that need to be applied to the ion lens elements. Finally, we have explained how we ensure that the central slice of the scattering distribution is being recorded, by tuning the delay between the laser pulse and camera trigger.

3. *“What structure information can the image provide?”*

We have clarified our discussion of the structural information that can be extracted from the experimental data, as well as the structural information that has been obtained in previous two-dimensional covariance-map imaging studies.

4. *“How to separate each channel on the total ion signal? What kinds of dissociation dynamics are involved?”*

We have expanded our discussion of the various fragmentation channels observed in our data, and the way in which these can be separated in the covariance mapping analysis. However, we note that this comment may have followed on from the referee’s belief that we were performing a pump-probe experiment. In any case, we hope that the way in which the various fragmentation channels can be identified is now clear.

We hope that we have addressed the referee’s comments to his/her satisfaction. His/her final comment requested additional measurements. Unfortunately, the nature of access to beam time means that we cannot go back and repeat these experiments at the Artemis laser facility, though we do have pump-probe experiments planned for future beam time. The facility is currently just starting up after a long shut down during which they moved premises, so access to beam time is relatively limited over the next year or so while they are getting back up to full capacity.

The referee also noted a missing journal name in one of the references (reference 21 of the original manuscript, though this will have changed in the revised manuscript). We have now included this.

Reviewer #3

1. *“The authors have concentrated on the straightforward two-body breakup $CF_3 - I$ and present nothing on the less obvious routes producing for instance F_2^+ . For a new technique, it would be essential to prove its validity beyond what is attainable through more conventional methods.”*

We have expanded our discussion of the various fragmentation channels of multiply charged CF_3I ions, and have included data for another channel to provide an additional example.

2. *“Even for the processes the authors have concentrated on ($CF_3^+ + I^+$ and $CF_3^+ + I_2^+$), the data shown are not quantitative, it is not possible to even estimate the degree of blurring of the velocity distributions and thence the attainable precision. All that is graphically shown is momentum conservation. This should be resolved.”*

This comment overlapped with comment no. 5 from reviewer #1. As noted above, we have now included a discussion of the current and possible future angular resolution of the technique.

3. *“In the (extremely short!) Discussion section, the authors say “With higher laser pulse energies, such that Coulomb explosion into atomic ions is achieved, the approach is able to provide direct information on three-dimensional molecular structures via the mapping from atomic positions to fragment ion velocities that occurs during the Coulomb explosion.”. This is a very interesting proposition but it remains to be demonstrated. One would first have to demonstrate that one can realistically break a polyatomic system into atomic ions and indicate the regime where this would happen (intense XFEL radiation?). References/previous work on this should be added. Also, the authors should give some indications on how they envisage the application of the covariance analysis in this multi-body situation (and how they would deal with the problem mentioned in comment 4 below).”*

We have shown in our previous work that we can break a relatively large polyatomic system into atomic ions and obtain structural information from velocity-map images of these Coulomb explosion products. We have expanded our discussion of this earlier work to make this clear. Relevant references were already provided in the original version of the manuscript. It has also been demonstrated previously that three-body covariances can be determined. The key result of the present manuscript is to extend covariance mapping from two dimensions to three dimensions through the introduction of 3D slice imaging in the detection step. When combined with three-body covariance mapping this will enable full 3D localisation of all atoms relative to each other. We have carried out extensive simulations of 3D covariance-map imaging to ensure its feasibility, but did not include these in the manuscript as they were carried out on a different molecular system. We thought that this would be confusing for the reader if presented in the context of our experiments on CF₃I, and therefore that it would be better to report the simulations in a separate publication.

4. *“On page 4 the authors say “These energies match the Coulomb potential energy of the relevant charges initially separated by a distance around 20% greater than the 2.144 Å C-I bond distance in neutral CF₃I, implying that the charge on the CF₃⁺ fragment is somewhat delocalised over the whole ion rather than being localised on the carbon atom.”. In my view this is a superficial hypothesis for a rather complex problem. There is ample previous literature on this general finding that observed kinetic energies resulting from Coulomb explosion processes are lower than those expected from Coulomb repulsion at equilibrium distances. There has been considerable effort to understand this, through different approaches (charge-resonance-enhanced ionization, ionization ignition, deviations from coulomb potentials,...). The authors should revise these, cite the relevant papers and discuss this effect properly. I include some of the relevant papers on this matter: T. Zuo et al., Phys. Rev. A, 52, R2511 (1995); C. Rose-Petrucci, et al., Phys. Rev. A: At.Mol. Opt. Phys., 55, 118 (1997); H. Liu et al., J. Chem. Phys. 126, 044316 (2007); M.E. Corrales, J. Phys. Chem. A 116, 2669 (2012). Also, the authors should indicate a route to extract structural information given this discrepancy with Coulomb energies expected for point charges at equilibrium distances. More generally, the Discussion section should be lengthened according to the ideas of comments 3 and 4 and an outlook for the applicability of the technique to larger systems should be provided.”*

The referee has made a very fair point in asking for further discussion of the information that can be obtained on bond lengths. We have explained that all chemical forces must be included when modelling Coulomb explosion processes, not just Coulomb repulsion,

and have included relevant references. We are currently working on developing our ability to predict Coulomb explosion images from various initial molecular structures through the use of on-the-fly trajectory simulations, with very promising early results.

5. *“I also believe that the paper should give proper credit to previous work on this matter. The 30-year old work by Vager et al., Science 244, 426 (1989) is a good example, but there are many other important ones (work by Stapelfeldt, Corkum, etc.)”*

We have added appropriate references as requested.

The referee also had some comments on the presentation of results and the Method section.

- (i) *“Figure 1 in page 3. The mass spectrum is extremely congested with spurious signals (CS₂, H₂O, N₂). The authors should either provide an explanation for this / explain why it has not been possible to clear the spectrum out, or redo the experiment in cleaner conditions.”*

We have added further explanation of the appearance of the time-of-flight mass spectrum presented in Figure 1, including a discussion of the origin of background peaks. That this was not included in the original manuscript was an oversight for which we apologise.

- (ii) *“-It is necessary to provide further information on the pulsed valve. The conditions of the molecular beam expansion and its repetition rate are not clear.”*

We have added further information on the pulsed valve and molecular beam.

- (iii) *“The paper would notably improve with more information regarding ion extraction conditions. The reader should have the relevant information for the slicing procedure, that is, the voltages applied and the definition of “slice”, that is, how the data are time-binned. The authors say “The electric field is tuned to retain the three-dimensional structure of the scattering (velocity) distribution for each of the ions”, but they do not explain how this “tuning” is performed.”*

We have added information on the velocity-mapping lens system, including voltages applied, slicing, and tuning of the ion lenses.

- (iv) *“There is also some lack of detail on gated voltages applied. In the description of the mass spectra the authors seem to indicate that the detector gain was turned down at the arrival times of the most intense peaks. The authors should indicate what degree of attenuation was employed, and with what time resolution. Also, I would expect a discontinuity in the MS shown in Figure 1 if the gain for H₂O is attenuated. Why is that not visible?”*

We have added details of the gating voltages used to time-gate the microchannel plate detector. The signal from ‘intense’ ions was not attenuated to zero, and the mass spectra shown are summed over several thousand acquisitions. A very small amount of time drift

during the acquisition, together with the rise and fall time of the gating electronics, tends to smooth out any very sudden step in the signal when the fields switch.

- (v) *“The authors should at least briefly describe the covariance analysis method as they apply it to their situation, even if they provide the relevant previous papers.”*

We have added a description of the covariance analysis to the end of the Methods section.

REVIEWERS' COMMENTS:

Reviewer #1 (Remarks to the Author):

The manuscript by Vallance et al. has been suitably improved, and my (and the other referees') comments were addressed properly.

I think the present version of the manuscript is publishable in Comm Chem.

Reviewer #2 (Remarks to the Author):

Following the paper revisions I support its publication.

Reviewer #3 (Remarks to the Author):

I believe that this manuscript has improved very substantially after revision. It is my view that the issues raised by the referees have been satisfactorily answered or corrected, including discussion on themes and important details that had been either overlooked or described too superficially in the first version. I also believe that the inclusion of data from an additional reaction channel is valuable and adds richness to the contribution. I also believe that this new version correctly gives credit to most relevant previous work.

The points that I raised in my first review have been positively addressed, the paper now represents a solid contribution and, in my view, it deserves publication in CommsChem.